# Neoadjuvant Chemotherapy in Breast Cancer: Evaluation of the Impact on Surgical Outcomes and Prognosis

**DOI:** 10.3390/cancers16132332

**Published:** 2024-06-26

**Authors:** Corrado Chiappa, Maltecca Greta, Leoni Miriam, Giuseppe Ietto, Davide Inversini, Andrea Ballabio, Alice Bonetti, Alberto Mangano, Rossana Gueli, Giulio Carcano, Francesca Angela Rovera

**Affiliations:** 1Senology Research Center, Department of Medicine and Innovation Technology (DiMIT), University of Insubria, 21100 Varese, Italy; corrado.doc@gmail.com (C.C.); gmaltecca@studenti.uninsubria.it (M.G.); mleoni3098@gmail.com (L.M.); alicebonetti1@gmail.com (A.B.); francesca.rovera@uninsubria.it (F.A.R.); 2Department of Medicine and Innovation Technology (DiMIT), University of Insubria, 21100 Varese, Italy; aballabio4@studenti.uninsubria.it (A.B.); giulio.carcano@uninsubria.it (G.C.); 3Division of General, Minimally Invasive and Robotic Surgery, University of Illinois at Chicago, 840 S Wood Street, Suite 435 E, Clinical Sciences Building, Chicago, IL 60612, USA; alberto.mangano@gmail.com; 4Oncology Unit, ASST-Sette Laghi, 21100 Varese, Italy; rossana.gueli@asst-settelaghi.it

**Keywords:** breast cancer, neoadjuvant chemotherapy, conservative surgery

## Abstract

**Simple Summary:**

Neoadjuvant treatment is an increasingly used treatment option for patients with both advanced and early-stage breast cancer to achieve downstaging and to improve prognosis. Therefore, it is important to evaluate its role in the multidisciplinary management of breast cancer patients and its effects on surgical outcomes and disease-free and overall survival. Achieving a pathologic complete response seems to improve disease-free and overall survival and even a partial response can be useful as an in-vivo chemosensitivity test for tailored adjuvant therapy. Clinical features, histology, and immunohistochemical findings play a role in achieving a pathological response. Therefore, they should be thoroughly investigated beforehand to better evaluate the treatment burden–benefits ratio and to predict the response.

**Abstract:**

The correlation between TNM staging and histology variations in a sample of patients who underwent neoadjuvant chemotherapy demonstrates a positive impact on both increasing conservative surgery and achieving pCR, resulting in better outcomes in terms of disease-free survival (DFS) and the risk of relapse. Benefits have also been highlighted in terms of cosmetic outcomes, postoperative complications, and psychological benefits. However, the overall outcomes must be evaluated according to the subtype and individual characteristics of the patients.

## 1. Introduction

This study aimed to evaluate TNM staging and histology variations in 261 patients with advanced or early-stage breast tumors undergoing neoadjuvant chemotherapy (NAC). The patients under consideration were treated over a 14-yeartime span at the S.S.D. Breast Unit, ASST Settelaghi of Varese. The secondary objectives were to evaluate the effect of NAC on both axillary and mammary disease downstaging and its impact on surgery (demolitive vs. conservative), the histopathological response, disease-free survival (DFS), and overall survival (OS). This study demonstrates promising results regarding NAC’s mass-reducing effect and its relation to conservative surgery and DFS.

## 2. Materials and Methods

The study sample was composed of 261 female patients aged between 25 and 84 years (average age at diagnosis of 51.8 years) with non-metastatic breast cancer diagnosed between January 2008 and December 2022. The patients underwent NAC and subsequent surgery. 

The sample was divided into two sub-samples, P1 (2008–2018) and P2 (2019–2022), due to substantial changes in clinical practice and international guidelines, such as the introduction of innovative regimens for both neoadjuvant and adjuvant chemotherapy. 

The exclusion criteria included metastasis findings during staging, neoadjuvant endocrine therapy(NEC), and incomplete data. 

Staging always included thoracic Rx, abdominal ultrasound, bone scintigraphy, and tumor marker CA15.3. CT and PET scans were administered when needed.

The age at diagnosis, a familiar history of breast or ovarian cancer, genetic mutations, comorbidities, and a personal history of cancer were taken into account. 

Patients underwent extensive imaging (breast ultrasound, mammography, and MRI) before, during, and after NAC, and the findings were recorded. Before undergoing NAC, a histological and/or cytological diagnosis, including immunohistochemical analysis, was made. Follow-up during NAC included monthly clinical evaluation and blood tests before every administration. 

The association between the variables and outcomes was tested using logistic models on a binary outcome basis. The estimates were calculated as odds ratios (ORs), with associated 95% confidence intervals (CIs).Overall survival (OS) and disease-free survival (DFS) were assessed using the Kaplan–Meier method. The association between the variables considered and death or recurrence was evaluated using separate Cox models. The estimates were expressed as hazard ratios (HRs) and related 95% confidence intervals. SAS 9.4 software was used for the analyses. The significance level was set at 0.05.

## 3. Demographics and Results

After dividing the sample into two time intervals (P1 and P2), a significant increase in the rate of both breast and axillary conservative surgery was observed (from 27.9% in P1 to 42.9% in P2, *p* = 0.0121, and from 19.8% in P1 to 42.2% in P2, *p* = 0.0002, respectively). This may be partially related to the changes observed at diagnosis in the immunohistochemical pattern of the neoplasia (and consequently its response to NAC), which saw an increase in HER2+ and triple negative tumors in P2. Similarly, there was a significant decrease in the rate of residual lymph-node-level invasive disease ypN ≥ 1 (from 41.4% in P1 to 28.0% in P2, *p* = 0.0007) and a percentage increase in ypT0 cases (from 21.6% in P1 to 32.3% in P2), resulting in a decrease in cases of tumors with ypT ≥ 1 on histological examination (from 64.8% in P1 to 55.3% in P2) (Figure 1 and Figure 2).

The mean size of the neoplasia at the time of diagnosis was 30.2 mm, whereas the imaging performed at the end of NAC and the final histological examination resulted in a mean value of 9.9 mm and a mean value of 10.1 mm (*p*-value < 0.0001), respectively.

At the time of diagnosis, the most frequent histotype was no special type (NST) infiltrating ductal carcinoma (251 cases), followed by infiltrating lobular carcinoma and less frequent histotypes, such as micropapillary, apocrine, and mucinous. Additionally, there were cases of metaplastic carcinomas and one case of neuroendocrine carcinoma. 

The most frequently observed grade at the time of diagnosis was G3, present in 175 of 272 patients. However, on the operative specimen, residual neoplasia was graded as G3 in 104 cases (*p*-value < 0.0001). 

The mean value of the Ki-67 proliferative index, indicating tumor growth rate, before therapy was 45.57%, whereas after treatment, it decreased to a mean value of 19.33% (*p*-value < 0.0001). In 196 of 272 (72.1%) patients, there was a decrease in Ki-67.

Finally, there was also a significant reduction in HER2 expression in HER2+ and Luminal B HER2+ tumors from 44.9% (122 of 272 patients) to 20.2% (55 of 272 patients) (*p*-value < 0.0001).

The type of breast surgery was significantly associated with periods, cT, and cN stages.

Out of a total of 272 cases, 100 patients (36.8%) underwent lumpectomy and 172 patients (63.2%) underwent mastectomy. However, comparing the rates of conservative surgery in periods P1 and P2, there was an increase in lumpectomy procedures from 27.9% to 42.9% (*p*-value = 0.0121).

This was due to both an increase in early-stage cases in P2,which would have undergone conservative surgery anyway, and an increase in cases that achieved downstaging.

A statistically significant difference (*p*-value = 0.0001) emerged between cN1 and cN0 tumors, with increased frequencies of demolitive surgery, defined as axillary lymph node dissection (ALND), among cN ≥ 1 patients (72.4%) compared to 49.5% of cN0 patients.

Examining the data separately by tumor subtype and cT staging, it appears that the choice rates between lumpectomy and mastectomy varied among Luminal A, Luminal B HER2-, HER2+, and triple negative subtypes (22.2% vs. 43.6% vs. 32.4% vs. 36.5%) and among different cT staging categories (*p* < 0.001 for the comparison between cT1, cT2, and cT3, 53.3% vs. 39.9% vs. 15.0%) [1].

However, in 39.9% of cT2 cases (63 of 158 patients) and in 15.5% of cT3 cases (3 of 20 patients), it was possible to change the initial indication from demolitive to conservative surgery. Specifically, 39% of Luminal B HER2+, 38% of HER2+, and 30.3% of triple-negative cT2 and cT3 tumors, initially ineligible for conservative surgery, became eligible after treatment due to the cytoreductive effects of NAC.

A total of 39.9% of initial cT2 tumors and 15.0% of cT3 tumors underwent surgical downstaging. A retrospective analysis of the final histologic examination in cases where conservative surgery had been opted for showed a pathologic complete response (pCR, ypT0) in 66.1% of cases. In parallel, 85.4% of cases found to be ypT ≥ 1 and ypT ≥ 2 had had indications for demolitive surgery.

The associated *p*-value (0.00012) indicates a significant difference.

The type of axillary surgical procedure (sentinel lymph node biopsy (SNLB), axillary lymphadenectomy/dissection (AD), or SNLB + AD) is significantly associated with the different study periods. There was a statistically significant increase in SLNB procedures (from 19.8% in P1 to 42.2% in P2) and a decrease in AD cases (from 79.3% in P1 and 56.5% in P2) (*p* = 0.0002), correlating with the increase in cN0 cases. 

Considering only cN0 patients, SLNB was performed in 86 cases with an increase from 71.0% in P1 to 82.1% in P2 (AD without SLNB in twenty cases and SLNB followed by AD in three cases). Among cN ≥ 1 tumors, axillary lymphadenectomy was the most frequently performed procedure (160 of 163 cases); however, in 80 out of 160 cases (50.0%), the final histologic examination results showed the absence of lymph node disease following downstaging obtained with NAC. In three cases, SLNB was performed, and the outcome was negative(Figure 3, Table 1 and Table 2).

The association between tumor subtype and the likelihood of performing SLNB versus AD was investigated using a logistic model, taking into account the variables cT, cN, and the observation period. However, the model’s results did not show a significant association between the different subtypes and a higher probability of SLNB. Regarding the pathologic complete response (pCR), which is the absence of residual invasive disease at the breast and lymph node level (ypT0 and ypN0, in patients who were previously assessed as cN+) or exclusive residual intraductal disease (ypT0/is ypN0), this was obtained in 36.4% of all cases analyzed (99 of 272 cases), with no statistically significant differences in the distribution of pathologic complete responses between the two periods.

On the other hand, there were statistically significant differences in the pathologic complete response rates among the different tumor subtypes (*p* < 0.001). The HER2+ subtype showed the highest response rate at 57.4%, followed by triple negative at 44.4%, Luminal B HER2+ at 35.3%, and Luminal B HER2− at 19.2%, whereas the Luminal A subtype exhibited the lowest response rate at only 11.1% (Figure 4, Table 3).

Another statistically significant difference emerged in the pathologic complete response rates between the cN categories of lymph node involvement. For cN0 patients, pCR occurred in 46.8% of cases, whereas in cN ≥ 1, pCR occurred in 29.4% of cases (*p* = 0.046).

There appeared to be no significant differences in the rate of pCR between the different stages of primary tumor size.

## 4. Discussion

Neoadjuvant chemotherapy treatment (NAC) aims to reduce tumor mass, ideally to the point of achieving a pCR, thereby reducing the local extent of the disease to enable conservative surgery.

During this study, significant changes were noted, including a substantial increase in indications for NAC for small cT1 and cT2 tumors without cN0 lymph node involvement, corresponding to early stage cTNM 1A and 2A (in agreement with the literature), as well as for advanced stage tumors to achieve a high clinical and pathologic response rate (pCR) and conversion to facilitate conservative surgery with disease-free margins [2].

Also, for cases of triple-negative breast cancer, NAC is recommended in the presence of cN+ and/or at least cT1c disease, sparing preoperative treatment in cT1a or cT1b cases in the absence of lymph node involvement.

In patients with HER2+ breast cancer with intermediate risk, defined based on cT1c and cN0 stage, as there is no consensus, the risk/benefit balance should be evaluated [3,4]. Although there is no documented difference in the literature in disease-free survival (DFS) or overall survival (OS) based on the pre- or postoperative timing of systemic therapy [5,6,7], there seems to be evidence that patients who achieved a pathologic complete response (pCR) had a significantly better prognosis than those who had residual disease. Furthermore, long-term data from these studies and subsequent meta-analyses have suggested that there are subpopulations of patients with certain tumor histotypes who experience greater benefits from treatment.

The results of this study are consistent with the literature on pCR (Luminal A and Luminal B HER2− (8.3%), Luminal B HER2+ (18.7%), HER2+ (38.9%), and triple negative (31.1%)) [8,9]. This study demonstrated a significant difference (*p* < 0.001) in the comparison of pCR frequencies between subtypes (Luminal A and Luminal B HER2− (18.4%), Luminal B HER2+ (35.3%), HER2+ (57.4%), and triple negative (44.4%)), confirming that the chance of pCR is significantly higher in groups of breast cancer patients with higher tumor aggressiveness, quantified by HER2 expression or the absence of estrogen receptors as a sign of cellular undifferentiation.

Globally, considering the population enrolled in this study, 36.4% pCR was obtained, which is higher than is reported in the literature [10], and the association between pCR and long-term outcome was stronger in patients with fast-growing tumors, including HER2+ and triple negative subtypes.

A recent meta-analysis on a large sample of patients demonstrated a survival advantage (DFS and OS) for patients who achieved pCR, particularly in HER2+ and triple negative cases. Additonally, the absence of additive advantage offered by post-surgical adjuvant treatment should pCR be achieved in the neoadjuvant phase was noted [11]. It is important to consider that the absence of pCR may guide the indication for adjuvant treatments, leading to a significant reduction in the risk of recurrence [1,12].

In terms of DFS, the sample analysis showed that those who do not achieve a pCR have a median six times higher risk of recurrence at five years than those who achieve a pathologic complete response (OR 6.88, 95% CI 2.42–19.56, *p*-value = 0.0003); however, an association model for overall survival (OS) could not be developed as no deaths were reported among the patients who achieved a pCR.

Another statistically significant result was the decrease in Ki-67 expression observed in 72.1% of cases between pre- and post-NAC values (*p* < 0.0001), whereas an increase was observed in 27.9% of cases. However, no statistically significant difference was measured between the decrease in Ki-67 and pCR (*p*-value = 1.000).

The results regarding the reduction of Ki-67, HER2, and grading align with the literature, which has demonstrated the value of Ki-67 as a prognostic and predictive biomarker in the setting of neoadjuvant therapy. However, some issues remain open, such as the need for better standardization of this parameter, the current impossibility of establishing a cut-off, and the limitation of its interpretation, which must consider the different variations among tumor subtypes and other prognostic factors [8,9,13,14,15,16].

Sentinel lymph node biopsy (SLNB) is the standard surgical practice after NAC for cN0 patients. When sentinel lymph nodes are not involved, the omission of further axillary lymph node dissection (AD) has been recommended. When there is evidence of sentinel lymph node involvement at SLNB, AD is indicated, as in the surgery-first approach [17,18].

In the study sample, 82% of the cN0 patients (89 of 109 cases) underwent SLNB, whereas(20 of 109 cases) underwent an emblée AD for suspicious cN lymph node status at FNAC staging, CT, or PET scan. In 11 out of 20 cases the outcome of histologic examination was positive for the presence of disease (ypN ≥ 1).

The St. Gallen Consensus Meeting in 2023 considered SLNB appropriate in cN = 1 cases, particularly when ≥3 sentinel lymph nodes, upon removal, were negative on extemporaneous histological examination [19,20].The scientific community has considered the omission of AD if the lymph node is affected by limited residual tumor, such as in the case of ITCs (isolated tumor cells) or micro metastases.

However, data in the literature are controversial about the prognostic significance of minimal residual tumor in sentinel lymph nodes post-NAC [21,22]. Currently, ongoing international trials are evaluating the possibility of omitting axillary lymphadenectomy, especially in cases of limited involvement, for example, in the prospective multicenter observational study Neonode 2, which aims to evaluate the impact of omitting axillary lymphadenectomy in the presence of micrometastatic post-NAC sentinel lymph nodes [23].

Other ongoing studies aim to compare the efficacy of axillary radiotherapy versus surgery to determine whether the former can replace the latter in the setting of chemotherapy-resistant lymph node disease [24,25].

The rate of no residual axillary disease in cN1 patients, reported in a recent meta-analysis, was 60% for HER2+ tumors, 45% for Luminal B HER2+ tumors, 48% for triple negative tumors, and 18% for Luminal A and B HER2− [26,27] tumors, which was comparable with the results of this study, where statistically significant differences (*p*-value = 0.0193) in terms of lymph node response (ypN0) emerged among cN1 patients with 67.9% (19 of 28 cases) for HER2+ tumors, 59.5% (22 of 37 cases) for Luminal B HER2+ tumors, 60.9% (14 of 23 cases) for triple negative tumors, and 21.6% (8 of 37 cases) for Luminal A and B HER2− tumors.

Considering NAC in terms of mass-reducing therapy, to facilitate conservative surgery in the cases that, at onset, would have had an indication for demolitive surgery, a surgical downstaging rate of 37.1% (66 of 178 cases) was obtained for cT ≥ 2 tumors, which is higher than the 16% reported in the literature [28,29].

A total of 38.6% (17 of 44 cases) of Luminal B HER2+ tumors, 37.5% (12 of 32 cases) of HER2+, 27.5% (11 of 40 cases) of triple negative tumors, and 34.2% (26 of 76 cases) of Luminal A and Luminal B HER2− cT2 and cT3 tumors that were initially ineligible for conservative surgery became eligible after treatment.

The number of lumpectomies was influenced by several factors beyond the initial tumor size, including residual breast disease, residual tumor volume to breast volume ratio, and the presence of BRCA1 and BRCA2 mutational status, consequently leading the surgeon to propose mastectomy despite the technical feasibility of quadrantectomy.

## 5. Conclusions

Neoadjuvant treatment represents an increasingly common approach for patients with not only advanced breast cancer but also patients with early-stage breast cancer.

The primary purpose of neoadjuvant treatment is not only to achieve downstaging of the neoplastic pathology in order to perform conservative surgery but also to improve patients’ prognosis. No significant associations were found between early-stage cT and the presence of a pathologic complete response, thus confirming the appropriateness of extending the indication of NAC for early-stage cT1-2 tumors. The positive impact of achieving a pCR in disease-free survival (DFS) was demonstrated with a significant decrease in the risk of relapse.

If pCR is not achieved, localized breast-only or lymph-node-only responses can still offer benefits to patients in terms of surgical downstaging, cosmetic outcomes, fewer postoperative complications, and psychological benefits, thus improving the quality of life of breast cancer patients.

The success of surgical treatment is related not only to the degree of response to neoadjuvant treatment but also to the proper planning of investigations both at the time of diagnosis and before surgery, necessitating a multidisciplinary approach (including a breast surgeon, oncologist physician, radiologist, radiation oncologist, pathologist, and plastic surgeon).

In conclusion, neoadjuvant chemotherapy represents a promising therapeutic strategy for the treatment of breast cancer. However, its efficacy must be evaluated according to the subtype and individual characteristics of patients. This analysis was beneficial in assessing the efficacy of neoadjuvant chemotherapy treatment and highlighted the changes that have emerged in clinical and surgical practice in recent years. Future research must continue to conduct in-depth studies on the neoadjuvant approach to optimize clinical and surgical downstaging outcomes at the breast and lymph node levels. Additionally, further investigation is needed to better understand the role of Ki-67 and other biomarkers in tailoring therapeutic strategies.

## Figures and Tables

**Figure 1 cancers-16-02332-f001:**
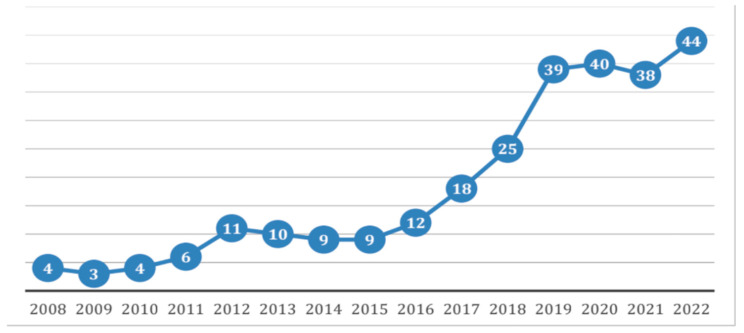
Case distribution from 2008 to 2022, shown as the number of patients per year.

**Figure 2 cancers-16-02332-f002:**
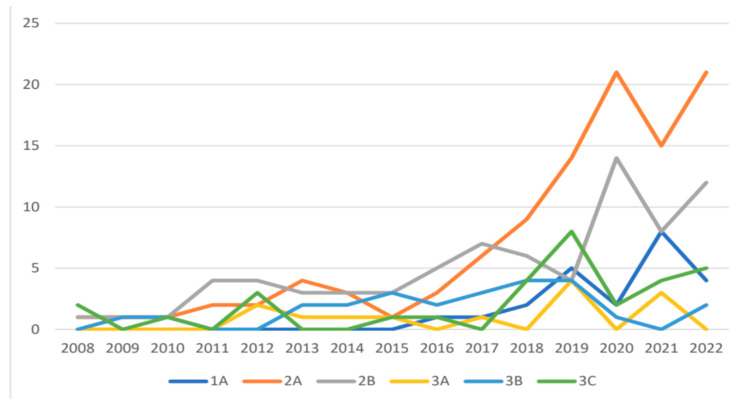
TNM distribution from 2008 to 2022, shown as the number of patients per year.

**Figure 3 cancers-16-02332-f003:**
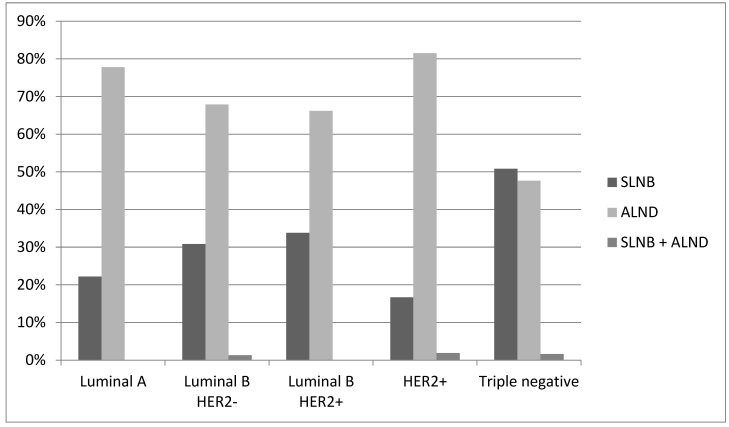
Post-NAC axillary surgery.

**Figure 4 cancers-16-02332-f004:**
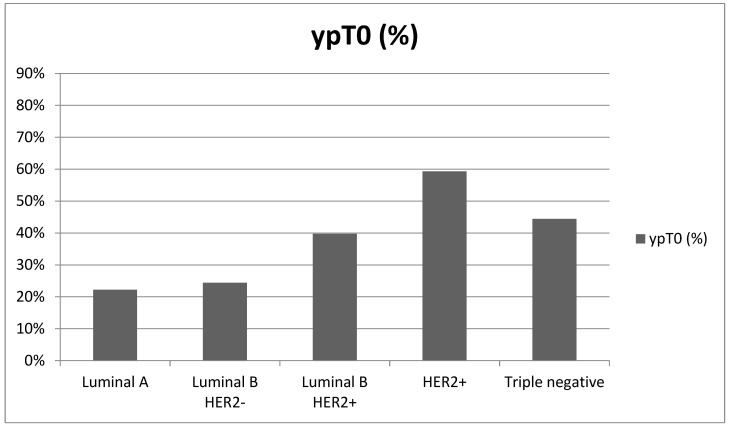
pCR based on histotype.

**Table 1 cancers-16-02332-t001:** Post-NAC axillary surgery.

		SNLB		ALND		SNLB + ALND		TOT	
		N°	%	N°	%	N°	%	N°	*p*-value
Patients		90	33.1%	179	65.8%	3	1.1%	272	
Period	P1	22	19.8%	88	79.3%	1	0.9%	111	0.0002
	P2	68	42.2%	91	56.5%	2	1.2%	161	
Histotype	Luminal A	2	22.2%	7	77.8%	0	0%	9	0.0054
	Luminal B HER2−	24	30.8%	53	67.9%	1	1.3%	78	
	Luminal B HER2+	23	33.8%	45	66.2%	0	0%	68	
	HER2+	9	16.7%	44	81.5%	1	1.9%	54	
	Triple negative	32	50.8%	30	47.6%	1	1.6%	63	
cN	cN0	86	78.9%	20	18.3%	3	2.8%	109	<0.0001
	cN1	4	3.2%	121	96.8%	0	0%	125	
	cN2	0	0%	6	100%	0	0%	6	
	cN3	0	0%	32	100%	0	0%	32	
cT	cT1	23	38.3%	37	61.7%	0	0%	60	<0.0001
	cT2	61	38.6%	95	60.1%	2	1.3%	158	
	cT3	6	30%	13	65%	1	5%	20	
	cT4	0	0%	31	100%	0	0%	31	
	cTx	0	0%	3	100%	0	0%	3	
cN0 patients		86	78.9%	20	18.3%	3	2.8%	109	
Period	P1	22	71%	8	25.8%	1	3.2%	31	0.4148
	P2	64	82.1%	13	12%	2	2.6%	78	
Histotype	Luminal A	2	66.7%	1	33.3%	0	0%	3	0.5479
	Luminal B HER2−	23	71.9%	8	25%	1	3.1%	32	
	Luminal B HER2+	22	88%	4	16%	0	0%	25	
	HER2+	9	64.3%	3	21.4%	1	7.1%	14	
	Triple negative	30	81.1%	4	10.8%	1	2.7%	37	
cT	cT1	23	92%	2	8%	0	0%	25	0.0115
	cT2	59	77.6%	14	18.4%	2	2.6%	76	
	cT3	6	85.7%	1	14.3%	1	14.3%	7	
	cT4	0	0%	3	100%	0	0%	3	
	cTx	0	0%	0	0%	0	0%	0	

**Table 2 cancers-16-02332-t002:** Post-NAC axillary surgery.

				CI 95%	
	SLNB vs. ALND	*p*-Value	OR	Inferior	Superior
Histotype	Luminal A	0.5096	3.11	0.11	90.54
	Luminal B HER2−		1.00 (ref)		
	Luminal B HER2+	0.4364	1.624	0.48	5.51
	HER2 +	0.519	0.64	0.17	2.47
	Triple negative	0.0957	2.91	0.83	10.25
cT	cT1		1.00 (ref)		
	cT2	0.7627	0.84	0.26	2.67
	cT3	0.3525	0.43	0.07	2.58
	cT4		Ns		
cN	cN0		1.00 (ref)		
	cN ≥ 1	<0.0001	0.01	0.00	0.02
Period	P1		1.00 (ref)		
	P2	0.2242	1.83	0.69	4.87

**Table 3 cancers-16-02332-t003:** Response to NAC based on histotype.

	ypT0-is		ypT ≥ 1		TOT	
	N°	%	N°	%	N°	*p*-Value
		39.7%	164	60.3%	272	
P1	108	34.2%	73	65.8%	111	0.1324
P2	38	43.5%	91	56.5%	161	
Luminal A	70	22.2%	7	77.8%	9	0.001
Luminal B HER2−	2	24.4%	59	75.8%	78	
Luminal B HER2+	19	39.7%	41	75.6%	68	
HER2+	27	59.3%	22	60.3%	54	
Triple negative	32	44.4%	35	55.6%	63	
cT1	33	55.0%	27	45.0%	60	0.0096
cT2	59	37.3%	99	62.7%	158	
cT3	3	15.0%	17	85.0%	20	
cT4	11	35.5%	20	64.5%	31	
cTx	2	36.7%	1	33.3%	3	

## Data Availability

Data are contained within the article.

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
