# Peer review of "Neoadjuvant Chemotherapy in Breast Cancer: Evaluation of the Impact on Surgical Outcomes and Prognosis"

_cancers, 2024, doi:10.3390/cancers16132332_

Round 1

Reviewer 1 Report

Comments and Suggestions for Authors

This is a nice summary of how neoadjuvant chemotherapy has changed surgical practices in a single center in Italy.

Some notes:

Data presentation is confusing.

Abbreviations should be spelled out before using them.  

Some parts are still in Italian and should be translated to English - see Tables.

Figure 1 - needs a y-axis and a better caption to understand what you are trying to show

Figure 2 - what do the numbers on the y-axis represent?

Page 3 - NST - what does that stand for?

Does demolitive surgery = mastectomy?  If so, would favor using the word mastectomy.

Page 3- "Examining the data separately by tumour subtype...." - are the percentages in that paragraph referring to lumpectomies?  That is not clear.  

The next paragraph gives too much detail in the parentheses making the sentences hard to follow.  Would combine that with the following paragraph and begin with the simple rates of changing surgery in cT2 and T3 patients.  Then would provide more of the details of histology.

Page 4 - what is BLNS and DA?  They are used without first being defined. It seems 2 different types of acronyms are being used for lymph node surgeries.  Would stick with SLNB and ALND, which are the more commonly utilized acronyms - change on the figures as well.

Page 4 - what is amble resection?

Page 6- would not refer to the cases that were cN0 as pathologic complete responses, as they may have been node negative the entire time (ie, before chemotherapy).  Would instead clarify that the rates of pCR in the breast was higher in patients who were cN0 versus cN+ prior to chemo.  The reference to Fig 4, Table 3 follows the statement about the differences seen in cN0 versus cN1 cases but these are not actually in the table/figure.  Would either add them or move the reference to them to a more appropriate location.

Discussion:

Last paragraph page 8: reference is made to p53 but this is not discussed at all previously in the manuscript - would remove

First paragraph, page 9 - would reference the Alliance trial 011202 which is looking at omission of ALND in patients who are still pN+ on SLNB after NAC

2nd paragraph - re-word emblee AD

Last paragraph of Discussion - re-word demolition surgery to mastectomy

Comments on the Quality of English Language

Abbreviations should be spelled out before using them.  

Some parts are still in Italian and should be translated to English - see Tables.

Some words are not appropriate in English and should be changed, as noted above

Author Response

All reviews have been addressed (see attachment)

Reviewer 2 Report

Comments and Suggestions for Authors

In the discussion they should concentrate more on the results, make it shorter.

No mention of the possibility of additional treatment if no pCR, specially in the Her 2 neg.

Author Response

Q: In the discussion they should concentrate more on the results, make it shorter.

A: The paragraph has been revised

Q: No mention of the possibility of additional treatment if no pCR, specially in the Her 2 neg.

A: The suggestion has been taken into account

Reviewer 3 Report

Comments and Suggestions for Authors

This is a presentation of the results of pre-operative [neo-adjuvant] chemotherapy given to hundreds of breast cancer patients in a single institution in Italy. Their results are quite similar to the published data and increase the confidence in this procedure.

They don't elaborate on the downside of the procedure like occasionally over treatment of small tumors and the possibility of disease progression in non-responding patients.

The legends of the figures and tables are too short and misleading

THE PAPER SHOULD BE PUBLISHED AS IT IS  A LONG-TERM STUDY THAT CONFIRMS AND ENCOURAGES THE USE OF THIS PROCEDURE.

Comments on the Quality of English Language

Require extensive revision and correction

Author Response

Q: They don't elaborate on the downside of the procedure like occasionally over treatment of small tumors and the possibility of disease progression in non-responding patients.

A: The suggestion has been taken into account

Q: The legends of the figures and tables are too short and misleading

A: Legends of the figures and tables have been revised